# Temperature Dependence of Raman Frequency Shift in SrWO$_4$ Crystal Studied by Lattice Dynamical Calculations

**Jun Suda** [1,*] **and Petr G. Zverev** [2]

1   Department of Electrical and Electronic Engineering, School of Engineering, Chukyo University,
    101-2 Yagoto Honmachi, Showa-ku, Nagoya-shi 466-8666, Japan
2   Prokhorov General Physics Institute Russian Academy of Sciences, Vavilov Str., 38, 119991 Moscow, Russia;
    zverev@lst.gpi.ru
*   Correspondence: suda@sist.chukyo-u.ac.jp

**Abstract:** The frequency shift of the Raman modes in strontium tungstate (SrWO$_4$) was investigated in the temperature range from 15 to 295 K. The experimental temperature dependence of the shift was analyzed using both the lattice dynamical calculations and the lattice perturbative approach. We found that the quartic anharmonic term of the first-order perturbation and the cubic term of the second-order perturbation, as well as the thermal expansion, contribute to the temperature shift of the highest-frequency A$_g$($\nu_1$) mode. The values of the temperature sensitivity of the frequency shift of the Raman modes at room temperature were measured, which is important for developing high-power crystalline Raman lasers and frequency shifters.

**Keywords:** Raman spectra; stimulated Raman scattering; lattice dynamical calculations; phonon dispersion; strontium tungstate; Raman laser

## 1. Introduction

Strontium tungstate (SrWO$_4$) crystal has been proposed as an excellent material for Raman lasers and frequency shifters due to the large Raman scattering cross section of the high-frequency A$_g$($\nu_1$) mode of the scheelite structure [1–3]. SrWO$_4$ crystal is used as an efficient crystal for stimulated Raman scattering (SRS) and for the development of crystalline nano- and picosecond Raman lasers [4–8]. Recently, it was shown that strong pulse shortening can be observed in synchronously pumped picosecond Raman lasers with single, multiple, and combined frequency shifts [9]. Therefore, the determination of the origin of anharmonicities of the SRS-active modes and the study of the temperature stability of their parameters are very important for the development of high-power solid-state Raman lasers.

It is necessary to note that the temperature dependences of the highest-frequency A$_g$($\nu_1$) Raman mode width are different in scheelite tungstate and molybdate crystals despite the same type of symmetric breathing vibrations [10–16]. Recently, we reported that the highest A$_g$($\nu_1$) Raman mode in SrWO$_4$ crystals exhibits large broadening with increasing temperature due to the cubic and dephasing terms of the anharmonic interaction, which was proven by both high-resolution Raman spectroscopy and lattice dynamical calculations [15]. We expect that the anharmonic effect plays an important role in the temperature dependence of Raman shifts and in the thermal expansion of the crystal. The aim of this research was to investigate the origin of the temperature shift of the Raman modes of SrWO$_4$ by applying the lattice dynamical method to the calculated phonon dispersion curves and phonon kinematics and to predict the stability of the SrWO$_4$ Raman laser parameters.

## 2. Materials and Methods

SrWO$_4$ single crystals were grown using the Czochralski technique. The sample of the crystal used in the present study was colorless and transparent, had optically polished faces cut perpendicular to the a and c axes, and was $5 \times 5 \times 6$ mm in size. A continuous-wave (CW) argon ion laser ($\lambda = 514.5$ nm), focused by a 35 cm lens, excited spontaneous Raman scattering in the sample, which was analyzed by a Spex-1403 double-spectrometer in a backwards scattering scheme. A cooled photomultiplier (PMT) controlled by a personal computer recorded the Raman scattering spectra. The spectral resolution of the whole system was sufficient to resolve Raman mode linewidths of less than 0.2 cm$^{-1}$. The spectral profiles of the Raman modes were well described by Lorentzians, which allowed us to extract the reliable values of their frequencies and linewidths. For temperature experiments, the sample was placed at the cold tip of a closed-cycle helium optical cryostat with temperature stabilization from 15 to 295 K with an accuracy of $\pm 1$ K. A chromel–gold thermocouple was fixed close to the sample. For low-temperature experiments, the laser intensity on the sample was reduced to less than 0.5 W.

## 3. Results

SrWO$_4$ crystal structure belongs to the C$_{4h}^6$ scheelite-type space group. Its primitive cell contains two formula units. The molecular ionic group [WO$_4$]$^{2-}$ with strong covalent bonds W–O is a peculiarity of the scheelite-type structure. Due to weak coupling between the ionic group and metal cation Sr$^{2+}$, the phonon modes in the spontaneous Raman spectra of SrWO$_4$ crystal can be divided into two groups, internal and external (Figure 1). The internal vibrons correspond to oscillations inside the [WO$_4$]$^{2-}$ molecular group with a motionless mass center. The external or lattice phonons correspond to the motion of the cation and the rigid molecular unit.

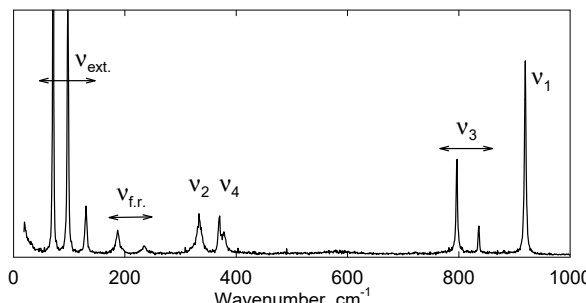

**Figure 1.** Spontaneous Raman spectrum of SrWO$_4$ crystal at room temperature.

[WO$_4$]$^{2-}$ tetrahedrons in free space have T$_d$ symmetry [1]. From factor group analysis, it follows that a [WO$_4$]$^{2-}$ free molecular ion has 3N = 15 degrees of freedom, which can be divided into four internal modes defined as $\nu_1$(A$_1$), $\nu_2$(E), $\nu_3$(F$_2$), and $\nu_4$(F$_2$), one free rotation mode $\nu_{f.r.}$(F$_1$), and one translation mode (F$_2$). When a [WO$_4$]$^{2-}$ ion is placed into the scheelite structure, its point symmetry reduces to S$_4$. This results in the splitting of all degenerate vibrations due to the crystal field effect [1]. According to the group theory calculations, the presence of two molecular groups in the primitive cell results in the presence of 26 different vibrations in the crystal structure: 3A$_g$ + 5 A$_u$ + 5 B$_g$ + 3 B$_u$ + 5 E$_g$ + 5 E$_u$. All the odd term vibrations (A$_g$, B$_g$, and E$_g$) are Raman active, the even term modes (4A$_u$ and 4E$_u$) can be registered only in the infrared spectra, three B$_u$ vibrations are silent modes, and one A$_u$ and one E$_u$ are acoustic vibrations.

The Raman laser SRS-active A$_g$($\nu_1$) vibronic mode was observed at 921 cm$^{-1}$, and intense E$_g$($\nu_3$) mode was found at 798 cm$^{-1}$, E$_g$($\nu_4$) at about 380 cm$^{-1}$, B$_g$($\nu_4$) at 380 cm$^{-1}$, and A$_g$+B$_g$($\nu_2$) at 335 cm$^{-1}$. The most intense external modes E$_g$($\nu_{ext.}$) and B$_g$($\nu_{ext.}$) were observed at 99 cm$^{-1}$ and 74 cm$^{-1}$, respectively. Hereinafter, we denote the frequencies of modes that were registered at room temperature. In the subsequent experiments, the spontaneous Raman spectra of the modes mentioned above were

measured with a high spectral resolution, which provided exact values of the frequency and linewidth of the Raman modes (Table 1).

**Table 1.** Frequency $\nu_R$, linewidth (FWHM) $\Delta\nu_R$, and Grüneisen parameters of the Raman modes in SrWO$_4$ crystal, as well as the slope coefficient of temperature dependencies $\nu_R(T)$ at room temperature.

| Raman Mode | $\nu_R$, cm$^{-1}$ | $\Delta\nu_R$, cm$^{-1}$ | $\gamma_R$ | $\partial\nu_R/\partial T$, (cm·grad)$^{-1} \times 10^{-3}$ |
|:---:|:---:|:---:|:---:|:---:|
| A$_g(\nu_1)$ | 921 | 3.02 | 0.15 | −5.67 |
| E$_g(\nu_3)$ | 798 | 3.28 | 0.19 | −7.0 |
| E$_g(\nu_4)$ | 380 | 6.61 | 0.14 | −5.3 |
| B$_g(\nu_4)$ | 372 | 3.71 | 0.16 | −6.0 |
| A$_g$ + B$_g(\nu_2)$ | 335 | 10.0 | 0.26 | −10.0 |
| E$_g(\nu_{ext.})$ | 100 | 2.48 | 0.35 | −13.2 |
| B$_g(\nu_{ext.})$ | 74 | 1.56 | 0.15 | −6.54 |

Experimental temperature-dependent Raman spectra of A$_g(\nu_1)$ internal and B$_g(\nu_{ext})$ lattice Raman modes in SrWO$_4$ crystal are presented in Figure 2. The frequency of A$_g(\nu_1)$ internal mode at the lowest temperature equaled 922.05 cm$^{-1}$. Its value was slightly increased to 922.33 cm$^{-1}$ when the crystal was heated up to 77 K, and then, its value reduced to 921.12 cm$^{-1}$ at 295 K. The frequency of B$_g(\nu_{ext})$ lattice mode continuously reduced from 74.13 to 73.29 cm$^{-1}$ with heating to room temperature. Similar temperature-dependent Raman spectra were measured for other modes. The frequency shift of these Raman modes with crystal temperature changing in the range from 15 to 295 K is presented in Figure 3. The frequencies of all modes decreased as temperature increased above 100 K. Here, we analyze the behavior of the A$_g(\nu_1)$ Raman mode frequency in more detail. The observed frequency shift $\Delta\nu_{obs}(T)$ is defined as the difference between the frequency obtained at the current temperature $\nu_{obs}(T)$ and the absolute frequency that is close to the value measured at 15 K $\nu_{obs}(15\,K)$, i.e., $\Delta\nu_{obs}(T) = \nu_{obs}(T) - \nu_{obs}(15\,K)$ (see Figure 4). It is known that the frequency shift $\Delta\nu_{obs}(T)$ observed at temperature T consists of the quasi-harmonic contribution from thermal expansion $\Delta\nu_0$ and the purely anharmonic contribution $\Delta\nu_A$ [17]:

$$\Delta\nu_{obs} = \Delta\nu_0 + \Delta\nu_A. \tag{1}$$

If we assume that the isothermal mode Grüneisen parameter $\gamma_{mode}$ is temperature-independent, we can estimate the frequency shift $\Delta\nu_0$ due to the thermal expansion as in [17]:

$$\Delta\nu_0 = -\nu(0)\gamma_{mode} \int_0^T \beta(T')dT' \tag{2}$$

where β(T) can be obtained by fitting the experimental values of the thermal expansion coefficients. The calculation of the mode Grüneisen parameters $\gamma_{mode}$ were done within the framework of density functional theory (DFT) using the Vienna ab initio simulation package (VASP) of which a detailed account can be found in [18] and references therein. The exchange energy was initially taken in the local density approximation (LDA) for tests and afterwards taken in the generalized gradient approximation (GGA) according to Perdrew–Bunrke–Ernzerhof (GGA-PBE) for a more exact calculation. The set of plane waves used extended up to a kinetic energy cutoff of 910 eV. This large cutoff was required to deal with the oxygen atoms with the projector augmented wave (PAW) scheme. The MonKhorst–Pack grid, used for integration over the Brillouin zone, ensured highly converged results (to about 1 meV per formula unit).

These calculated values of the mode Grüneisen parameter $\gamma_{mode}$ are in good agreement with those obtained under high-pressure experiments on BaWO$_4$ crystal by Manjón [19]. We calculated the frequency shift $\Delta\nu_0$ due to the thermal expansion using the volumetric thermal expansion coefficient in SrWO$_4$ [20,21]. For both A$_g(\nu_1)$ and B$_g(\nu_{ext.})$ in SrWO$_4$, we obtained a value of 0.15, respectively.

The calculated values for the other Raman modes are presented in Table 1. Similar smooth temperature dependencies for the other Raman modes in $SrWO_4$ at normal pressure confirmed its sheelite crystal structure and the absence of any phase transition below 295 K which correspond to the results obtained by Fan et al. [21]. Because the Raman linewidth was explained by the anharmonic cubic term [15], we supposed that the frequency shift was due to the true anharmonic effect as well:

$$\Delta \nu_A \cong \Delta \nu_A^{(3)} + \Delta \nu_A^{(4a)} \tag{3}$$

where the cubic and quartic terms are given as

$$\Delta \nu_A^{(3)}(j; \nu_R) = -\frac{h\nu_R}{64N} |C_3|^2 \sum_q \sum_{j_1, j_2} \nu_{j_1}(q) \nu_{j_2}(-q) \wp \left[ \frac{n_{j_1}(q) + n_{j_2}(q) + 1}{\nu_R + \nu_{j_1}(q) + \nu_{j_2}(q)} \right.$$
$$\left. - \frac{n_{j_1}(q) + n_{j_2}(q) + 1}{\nu_R - \nu_{j_1}(q) - \nu_{j_2}(q)} + \frac{n_{j_1}(q) - n_{j_2}(q)}{\nu_R - \nu_{j_1}(q) + \nu_{j_2}(q)} - \frac{n_{j_1}(q) - n_{j_2}(q)}{\nu_R + \nu_{j_1}(q) - \nu_{j_2}(q)} \right] \tag{4}$$

and

$$\Delta \nu_A^{(4a)}(j; \nu_R) = \frac{h\nu_R}{8N} C_4' \sum_{q, j_1} \nu_{j_1}(-q) \left( n_{j_1}(q) + \frac{1}{2} \right) \tag{5}$$

where $\wp$, N, and $n_j(q) = \left[ exp\left( \frac{h\nu_j}{kT} \right) - 1 \right]^{-1}$ are the principal value, the number of atoms in the unit cell, and the phonon occupation number for $\nu_j$ Raman mode, respectively. On the right-hand side (rhs) of Equation (4), $\Delta \nu_A^{(3)}$ makes a negative contribution to the line shift, while $\Delta \nu_A^{(4q)}$ in Equation (5) has a positive effect on the line shift.

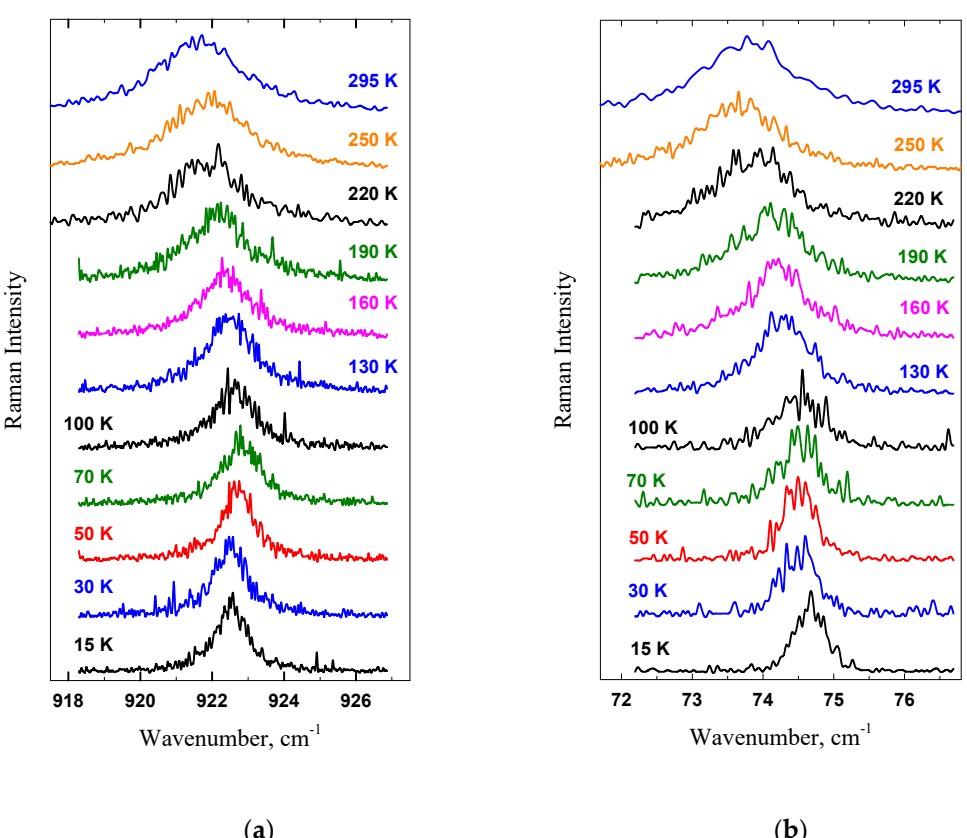

**Figure 2.** Temperature-dependent Raman spectra of $A_g(\nu_1)$ internal Raman mode (**a**) and $B_g(\nu_{ext})$ lattice mode (**b**) in $SrWO_4$ crystal.

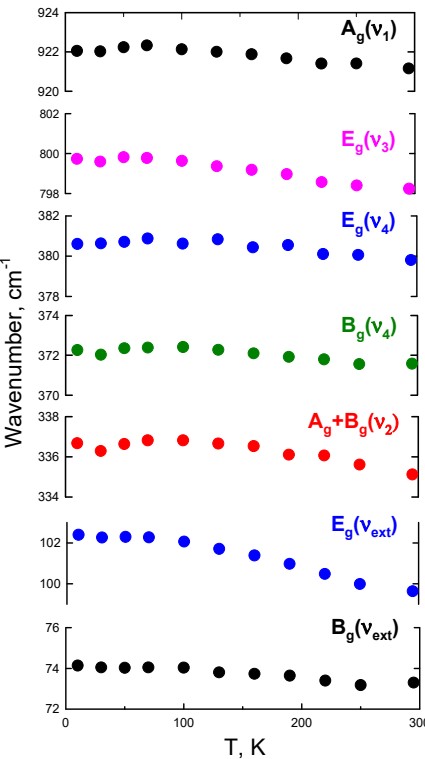

**Figure 3.** Temperature dependences of the frequencies of the Raman modes in SrWO$_4$ crystal.

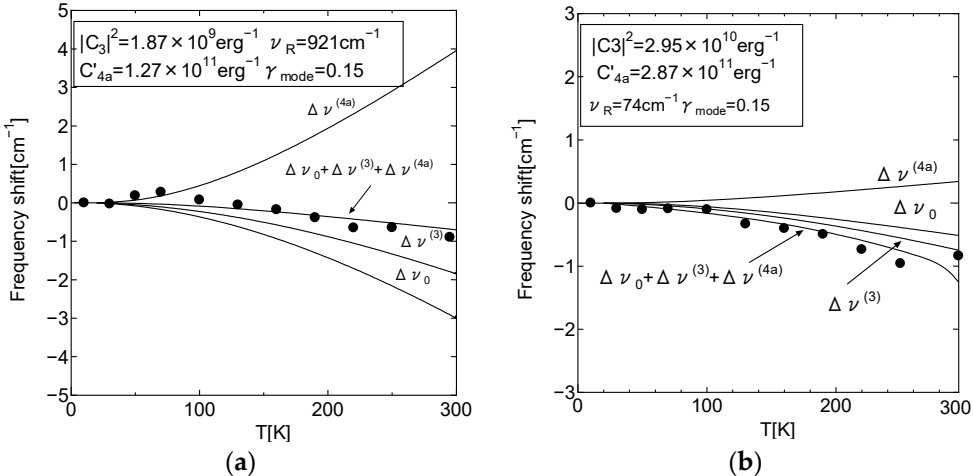

**Figure 4.** Temperature dependence of the frequency shift of the (**a**) internal A$_g$($\nu_1$) and the (**b**) external B$_g$($\nu_{ext}$) modes in SrWO$_4$ crystal. The experimental data are presented by filled circles, while the calculated dependences are presented by solid lines.

The simplicity of the first-order shift is partly a consequence of the momentum transfer in Raman spectroscopy. In the calculation of the rhs of Equations (4) and (5), the summation over q was performed using the phonon dispersion relations obtained from lattice dynamical calculations; the details are described in [15]. The reciprocal-space summation was performed using a linear tetrahedron method on a mesh of q vectors ($8 \times 8 \times 8$ grid mesh) in the first Brillouin zone, giving good convergence. Using Equations (1)–(5), for each Raman mode, both its frequency shift and broadening were fitted simultaneously with the two parameters $|C_3|^2$ and $C'_4$. The rhs of Equations (4) and (5) are proportional to the two anharmonic parameters $|C_3|^2$ and $C'_4$ [11]. In the calculation of $\Delta\nu_A^{(3)}$ and $\Delta\nu_A^{(4q)}$ without those parameters, the summation over q was carried out using the phonon dispersion relations obtained from lattice dynamical calculations [12]. We fitted the value of C$_4$′ in

Equations (4) and (5) to obtain the experimental value of the linewidth at 200 K. For $A_g(\nu_1)$ and $B_g(\nu_{ext})$ modes, we calculated the value of $C'_4$ to be $1.27 \times 10^{11}$ erg$^{-1}$ and $2.87 \times 10^{11}$ erg$^{-1}$ by using the same values of $|C_3|^2$ [12]. The calculated values of $\Delta\nu_A$ for both $A_g(\nu_1)$ and $B_g(\nu_{ext})$ modes were in good agreement with the observed ones in a temperature range of 15–295 K, as shown in Figure 4. The value of the quartic contribution $\Delta\nu_A{}^{(4q)}$ had approximately the same order as $\Delta\nu_A{}^{(3)}$, consistent with the fact that $\Delta\nu_A{}^{(4q)}$ and $\Delta\nu_A{}^{(3)}$ were both the leading order terms for the anharmonic frequency shift of $A_g(\nu_1)$ mode. The temperature behavior of the $A_g(\nu_1)$ mode frequency was unusual, i.e., the net anharmonic shift was positive without the frequency shift $\Delta\nu_0$ due to the thermal expansion, which indicates that the quartic contribution to the shift was larger than those of the cubic.

The linewidth of $A_g(\nu_1)$ mode was found to be narrower than those of other modes [15], because the cubic term was restricted by a wide energy band gap between $\nu_3$ and $\nu_4$ Raman modes. The effect due to the quartic term showed a positive effect on the frequency shift. Therefore, the negative effect due to both thermal expansion and the cubic term was cancelled by the quartic term at low temperatures for $A_g(\nu_1)$ internal mode and prevailed at higher temperatures (Figure 4a). One can see that the frequency shift of $B_g(\nu_{ext})$ external mode is well described by the cubic effect and thermal expansion effect, because the quartic effect showed smaller impact in the whole temperature range (Figure 4b).

## 4. Discussion

The Raman frequency shift of the $A_g(\nu_1)$ mode in SrWO$_4$ was measured in the temperature range 15–295 K. The temperature dependence of the Raman shift of this mode was analyzed using both lattice dynamical calculations and the lattice perturbative approach. The calculated results for $A_g(\nu_1)$ describe well the experimental data on frequency shift and line broadening in the temperature range 15–295 K. We found that the quartic anharmonic term of the first-order perturbation, as well as the cubic term of the second-order perturbation and thermal expansion, contributed to the temperature dependence of the frequency shift of the Raman modes. However the behavior of the frequency shift of the $A_g(\nu_1)$ mode was unusual. At low temperature it showed a positive net shift anharmonic shift, which indicated that the positive quartic term $\Delta\nu_A^{(4a)}$ was larger than the cubic one $\Delta\nu_A^{(3)}$, because the cubic term is restricted by a wide energy gap in SrWO$_4$ crystal. For high-energy internal modes, the broadening was mostly due to the splitting process with the excitation of two phonons with lower frequencies [15]. This indicates the presence of a large phonon band gap in the phonon density of states, which is responsible for the difference in the anharmonic effects on the Raman shift of high- and low-frequency modes.

The temperature sensitivity of the Raman frequency and Grüneisen parameters of the modes in the vicinity of room temperature are presented in Table 1. The symmetrical $A_g(\nu_1)$ mode has the lowest sensitivity to the crystal temperature. The overlapping of two symmetric bending modes $(A_g + B_g)(\nu_2)$ (the widest Raman line) can be used for SRS of picosecond pulses due to a short relaxation time [9], but it is rather sensitive to the crystal temperature. The obtained results allow one to determine the temperature sensitivity of the Raman mode frequency, which is important for developing high-power Raman lasers and frequency converters based on SrWO$_4$ crystal and for predicting the exact value of the Stokes radiation wavelength in such devices.

**Author Contributions:** J.S. performed the lattice dynamic calculations; P.G.Z. conducted the spontaneous Raman experiments; and both authors discussed the results, wrote the text, and substantially contributed to the manuscript.

**Funding:** This research was funded by the collaborative research program 2013, Information Initiative Center, Hokkaido University and by a Grant-in-Aid for Scientific Research (C) from the Ministry of Education, Science and Culture of Japan, Grant No. 20540322.

**Conflicts of Interest:** The authors declare no conflict of interest.

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
