# Peer review of "Temperature Dependence of Raman Frequency Shift in SrWO4 Crystal Studied by Lattice Dynamical Calculations"

_crystals, doi:10.3390/cryst9040197_

Round 1
Reviewer 1 Report
The main article focus is about the frequency shift of Raman modes in Strontium tungstate which was investigated in the temperature range from 15 to 295 K. The study is experimental together with calculations.
The paper must be rewritten because it is badly organized and a lot of information is missed. The introduction miss description of the problem which the authors want to study. The Discussion part represents the results and there are lack of conclusion part.
The experimental part is very weakly described. The calculations aren’t described at all: There are said they used VASP in Results section. Which was used pseudopotential? How were run calculations? All this information must be described in separate section. Thus the “Materials and Methods” should be divided into experimental and calculation parts. There should be represented how it looks like the Strontium tungstate structure in order to make the article more readable. Moreover, the equations (4) and (5) are represented in very poor quality. The paper is very messy and it is difficult understand the importance of the results. It cannot be published in present form.
Author Response
Response to Reviewer 1 Comments
We appreciate the reviewers’ very helpful and constructive comments on our manuscript. We checked the manuscript` and strongly improved English grammar and spelling throughout the text.
Following the remarks and suggestions, we have revised the manuscript as highlighted in the text. Our responses to the single comments are given as follows:
Point 1:
The introduction miss description of the problem which the authors want to study.
Response 1:
We added (lines 40-42):
The aim of this research was to investigate the origin of the temperature shift of the Raman modes of SrWO4 by applying the lattice dynamical method to the calculated phonon dispersion curves and phonon kinematics and predict the stability of SrWO4 Raman laser parameters.
Point 2:
The Discussion part represents the results and there are lack of conclusion part.
Response 2:
We rewrote (lines 172-175)
The obtained results allow one to determine the temperature sensitivity of the Raman mode frequency which is important for developing high-power Raman lasers and frequency converters based on SrWO4 crystal and for predicting the exact value of the Stokes radiation wavelength in such devices.
Point 3:
The experimental part is very weakly described.
Response 3:
We added additional information in the “Materials and Methods” section (lines 46-49 and 53-55):
Cw argon ion laser (λ = 514.5 nm) focused by 35 cm lens excitated spontaneous Raman scattering in the sample which was analyzed by a Spex-1403 double-spectrometer in a backwards scattering scheme. A cooled PMT controlled by a personal computer recorded Raman scattering specta.
and
For temperature experiments, the sample was placed at the cold tip of a closed-cycle helium optical cryostat with temperature stabilization from 15 to 295 K with the accuracy of ±1 degree. The chromel-gold thermo-couple was fixed close to the sample. For low-temperature experiments, the laser intensity at the sample was reduced to less than 0.5 W.
Point 4:
The calculations aren’t described at all: There are said they used VASP in Results section. Which was used pseudopotential? How were run calculations? All this information must be described in separate section. Thus the “Materials and Methods” should be divided into experimental and calculation parts.
The details of the calculations, parameters of the crystal lattice, force constants, detailed description of the calculation procedure are described in our previous publication [15].
We added (lines 96-105).
Response 4:
The calculation of the mode Grüneisen parameters γmode were done within framework of the density functional theory (DFT) using the Vienna ab initio simulation package (VASP) of which a detailed account can be found in [18] and references therein. The exchange energy was initially taken in the local density approximation (LDA) for tests and afterwards taken in the generalized gradient approximation (GGA) according to Perdrew-Bunrke-Ernzerhof (GGA-PBE) for more exact calculation. The set of plane waves used extended up to a kinetic energy cutoff of 910 eV. This large cutoff was required to deal with the oxygen atoms with the projector augmented wave (PAW) scheme. The MonKhorst-Pack grid used for integration over Brillouin-zone ensured highly converged results (to about 1 meV) per formula unit).
We added (lines 125-126).
In the calculation of the rhs of eqs. (4) and (5), the summation over q was performed using the phonon-dispersion relations obtained from lattice dynamical calculations, the details are described in ref. [15].
Point 5::
There should be represented how it looks like the Strontium tungstate structure in order to make the article more readable.
Response 5:
The strontium tungstate structure belongs to the C4h6 scheelite-type space group. There are many pictures with scheelite-type structure in the literature. Its drawing is presented in our previous publication [15].
Point 6:
Moreover, the equations (4) and (5) are represented in very poor quality.
Response 6:
We retyped these equations (lines 109-112).
Reviewer 2 Report
The draft has studied the frequency shift of Raman mode in SrWO4 in the temperature range from 15 to 295 K. The authors discussed that the shift resulted from the anharmonic contribution and the thermal expansion. The theoretical results were also compared with the experimental results. The draft is organized well and the language is fluent and precise. I think the work is interesting for the readers. I recommend the paper for publication, however, I still have several concerns.
.
First, in Fig. 1, there are some unassigned Raman peaks, what are these peaks?
Secondly, in Fig. 3, I think the authors should re-organized the figure and caption. Also, I think C4a should be C’4a. Moreover, when the temperature is around 40 and 60 K, the experimental results are most likely along \delta V(4a), any physical explanation for it?
Author Response
Response to Reviewer 2 Comments
We appreciate the reviewers’ very helpful and constructive comments on our manuscript. We checked the manuscript` and strongly improved English grammar and spelling throughout the text.
Following the remarks and suggestions, we have revised the manuscript as highlighted in the text. Our responses to the single comments are given as follows:
Point 1:
First, in Fig. 1, there are some unassigned Raman peaks, what are these peaks?
Response 1:
These peaks belong to the same type of modes n3, nf.r. or next. .
We added arrows in Fig.1 to denote those type of modes for peaks respectively.
Point 2:
Secondly, in Fig. 3, I think the authors should re-organized the figure and caption. Also, I think C4a should be C’4a. Moreover, when the temperature is around 40 and 60 K, the experimental results are most likely along \delta V(4a), any physical explanation for it?
Response 2:
We changed C’4a to C4a in Fig. 3. When the temperature was around 40 and 60 K, the experimental values showed small difference that was close to zero. This is due to the fact that the impact of thermal expansion (coefficient ΙC3Ι2 ) is much less for internal phonon Ag(n1) than that for external Bg(next) (Fig.3).
We added (lines 142-147):
The effect due to quartic term shows positive effect on frequency shift. Therefore, the negative effect due to both thermal expansion and the cubic term was cancelled by the quartic term at low temperatures for Ag(n1) internal mode and prevailed at higher temperatures (Fig. 3a). One can see that the frequency shift of Bg(next) external mode is well described by the cubic effect and thermal expansion effect because the quartic effect shows smaller impact in the whole temperature range (Fig.3b).
Reviewer 3 Report
In the present paper, authors measured temperature dependence of Raman active modes in SrWO4 and analyse anharmonic effect of the modes linewidth.
All the study is based on low temperature data collection and anharmonic mode fitting and I have to admit that in the present paper, due to lack of information, I was not able to start my review.
First, low temperature Raman spectroscopy using a cold finger cryostat requires special care to avoid/estimate laser overheating. Two options: (i) Working with very low power or (ii) measure the real temperature with stokes-anti-stokes ratio that should be possible in the present case at quite low temperatures using the two intense low energy modes below 100 cm-1. In the present version of the manuscript, no method on how the real temperature is estimated is described. The power, spot size, power density of the laser is not given. The authors should give any argument that show they take care about these aspects.
Another point is the analysis made about peak position and peak shape. Authors should show row data at low temperatures and corresponding fits from which parameters are extracted.
Once these two major elements are provided, paper may be considered for review.
Author Response
Response to Reviewer 3 Comments
We appreciate the reviewers’ very helpful and constructive comments on our manuscript. We checked the manuscript` and strongly improved English grammar and spelling throughout the text.
Following the remarks and suggestions, we have revised the manuscript as highlighted in the text. Our responses to the single comments are given as follows:
Point 1:
In the present paper, authors measured temperature dependence of Raman active modes in SrWO4 and analyse anharmonic effect of the modes linewidth.
All the study is based on low temperature data collection and anharmonic mode fitting and I have to admit that in the present paper, due to lack of information, I was not able to start my review.
First, low temperature Raman spectroscopy using a cold finger cryostat requires special care to avoid/estimate laser overheating. Two options: (i) Working with very low power or (ii) measure the real temperature with stokes-anti-stokes ratio that should be possible in the present case at quite low temperatures using the two intense low energy modes below 100 cm-1. In the present version of the manuscript, no method on how the real temperature is estimated is described. The power, spot size, power density of the laser is not given. The authors should give any argument that show they take care about these aspects.
Response 1:
The excitation radiation didn’t heat the sample noticeably. In our experiments we used transparent sample, the focusing of the excitation radiation was with the long focal length lens. The chromel-gold thermo-couple was fixed close to the sample. So we think that the temperature of the sample was measured with the accuracy of ±1 degree.
We made special precautions for low-temperature experiments. Narrow and intense Stokes lines allowed to measure the spontaneous Raman spectrum at very low pump energy. The laser intensity at the sample was reduced to less than 0.5 W.
We added and corrected the description of experiment (lines 46-49 and 53-55):
Cw argon ion laser (λ = 514.5 nm) focused by 35 cm lens excitated spontaneous Raman scattering in the sample which was analyzed by a Spex-1403 double-spectrometer in a backwards scattering scheme. A cooled PMT controlled by a personal computer recorded Raman scattering specta.
and
For temperature experiments, the sample was placed at the cold tip of a closed-cycle helium optical cryostat with temperature stabilization from 15 to 295 K with the accuracy of ±1 degree. The chromel-gold thermo-couple was fixed close to the sample. For low-temperature experiments, the laser intensity at the sample was reduced to less than 0.5 W.
Point 2:
Another point is the analysis made about peak position and peak shape. Authors should show row data at low temperatures and corresponding fits from which parameters are extracted.
Response 2:
The raw data of spontaneous Raman spectra of the internal Ag(n1) and lattice Bg(next) Raman modes in SrWO4 crystal are presented in our earlier publication [15].
Reviewer 4 Report
The authors have apparently split this work into two papers, one of which has been published already on the temperature-dependence of the Raman linewidth. Here the authors present their results and discussion on the temperature-dependent Raman frequency. This is unusual in the literature, where both frequency and linewidth are usually presented together. However, since the linewidth has already been published, the authors can improve the present manuscript by addressing the following -
Can the authors comment on what one expects with regards to anharmonicity along the various crystalline directions of SrWO4? If there are differences, they could be observable through polarized Raman analyis.
In the discussion the authors mention a difference in the phonon gaps between high- and low-energy phonon modes. Here they only present results for a single (high-frequency) mode. These differences can be shown more clearly by also presenting temperature-dependent shifts of low-frequency Raman modes. This shouldn't be difficult since they already have the data.
There are minor grammatical errors throughout the manuscript that should be corrected. A notable error is on line 131, Page 4. The word "shift" is misspelt.
Author Response
Response to Reviewer 4 Comments
We appreciate the reviewers’ very helpful and constructive comments on our manuscript. We checked the manuscript` and strongly improved English grammar and spelling throughout the text.
Following the remarks and suggestions, we have revised the manuscript as highlighted in the text. Our responses to the single comments are given as follows:
Point 1:
Can the authors comment on what one expects with regards to anharmonicity along the various crystalline directions of SrWO4? If there are differences, they could be observable through polarized Raman analysis.
Response 1:
SrWO4 crystal has weak anisotropy of physical properties due to scheelite-type structure. This anisotropy results in slight change of the value of Raman peak cross section [16]. However the frequency and linewidth of internal and lattice Raman modes are not changed. Polarized Raman spectroscopy can hardly give us any additional information about the frequency and the linewidth of SRS-active Raman mode.
The mode assignment of phonon frequencies in this type of crystals has been done by polarized Raman analysis [1,2]. This can be done by phonon calculations also. We have reported about this in our ref. [15].
Point 2:
In the discussion the authors mention a difference in the phonon gaps between high- and low-energy phonon modes. Here they only present results for a single (high-frequency) mode. These differences can be shown more clearly by also presenting temperature-dependent shifts of low-frequency Raman modes. This shouldn't be difficult since they already have the data.
Response 2:
We added Fig.3b that describes frequency shift of Bg mode (74 cm-1) (lines 148-151).
We added the following description (line 142-147) :
The effect due to quartic term shows positive effect on frequency shift. Therefore, the negative effect due to both thermal expansion and the cubic term was cancelled by the quartic term at low temperatures for Ag(n1) internal mode and prevailed at higher temperatures (Fig. 3,a). One can see that the frequency shift of Bg(next) external mode is well described by the cubic effect and thermal expansion effect because the quartic effect shows smaller impact in the whole temperature range (Fig.3b).
We revised the Fig.3 caption:
Figure 3. Temperature dependence of the frequency shift of the (a) internal Ag(n1) and the (b) external Bg(next) modes in SrWO4 crystal.
In references, we corrected refs.18 and 19.
Point 3:
There are minor grammatical errors throughout the manuscript that should be corrected. A notable error is on line 131, Page 4. The word "shift" is misspelt.
Response 3:
We checked the text for grammar and spelling mistakes and strongly improved the manuscript.
Round 2
Reviewer 1 Report
It can be accepted in present form. However, the better way would not only reference everything to [15] but give readers more details in this paper also.
Author Response
We appreciate the reviewers’ time and constructive comments on our manuscript.
Following the suggestions, we have further improved our manuscript as highlighted in the text. Our responses to the single comments are given as follows:
Point 1:
It can be accepted in present form. However, the better way would not only reference everything to [15] but give readers more details in this paper also.
Response 1:
Thank you for your positive decision. We added Figure 2 with experimental data to demonstrate temperature-dependent Raman spectra ofAg(n1) internal and Bg(next) lattice Raman modes in SrWO4 crystal that were analyzed in the subsequent parts of the manuscript. We changed figure numbering and added the description of Fig.2 in the text (lines 88-93):
Experimental temperature-dependent Raman spectra of Ag(n1) internal and Bg(next) lattice Raman modes in SrWO4 crystal are presented in Fig. 2. The frequency of Ag(n1) internal mode at the lowest temperature equaled to 922.05 cm-1. Its value was slightly increased to 922.33 cm-1 with heating the crystal up to 77K and then reduced to 921.12 cm-1 at 295 K. The frequency of Bg(next) lattice mode continuously reduced from 74.13 cm-1 to 73.29 cm-1 with heating to room temperature. Similar temperature-dependent Raman spectra were measured for other modes.
Reviewer 3 Report
Despite the paper writing was highly improved I think that the paper cannot be accepted in the present form.
All the study is based on fine details about 920 cm-1 peak position temperature dependence.
First of all, I insist the fact that temperature dependence of Raman signal which is the starting point of this study should be included in the present paper even if it was published in a former article. This would make the reading smoother.
Second, determination of the temperature is crucial for that work as slight modification of the temperature linearity in Fig. 3 may completely change the conclusions, I am not convinced by the answer of authors concerning the temperature control. They say they measure with very low laser power (0.5 W) but without telling the laser spot size… We are used to measure on transparent materials with a power density below 100-1000 kW/m2 to secure absence of sample heating in cryostats without exchange gas. This limit of course depends a lot on the material itself but could the authors state about power density in their experiment? And at least give this value?
Finally, in the ref. 15 an additional temperature at 295K was measured. Is there any reason why it is absent in Fig. 3 ?
Author Response
Response to Reviewer 3 Comments
We appreciate the reviewers’ time and constructive comments on our manuscript.
Following the suggestions, we have further improved our manuscript as highlighted in the text. Our responses to the single comments are given as follows:
Point 1:
First of all, I insist the fact that temperature dependence of Raman signal which is the starting point of this study should be included in the present paper even if it was published in a former article. This would make the reading smoother.
Response 1:
We added Figure 2 with experimental data to demonstrate temperature-dependent Raman spectra of Ag(1) internal and Bg(ext) lattice Raman modes in SrWO4 crystal that were analyzed in the subsequent parts of the manuscript. We changed figure numbering and added the description of Fig.2 in the text (lines 88-93):
Experimental temperature-dependent Raman spectra of Ag(n1) internal and Bg(next) lattice Raman modes in SrWO4 crystal are presented in Fig. 2. The frequency of Ag(n1) internal mode at the lowest temperature equaled to 922.05 cm-1. Its value was slightly increased to 922.33 cm-1 with heating the crystal up to 77K and then reduced to 921.12 cm-1 at 295 K. The frequency of Bg(next) lattice mode continuously reduced from 74.13 cm-1 to 73.29 cm-1 with heating to room temperature. Similar temperature-dependent Raman spectra were measured for other modes.
Point 2:
Second, determination of the temperature is crucial for that work as slight modification of the temperature linearity in Fig. 3 may completely change the conclusions, I am not convinced by the answer of authors concerning the temperature control. They say they measure with very low laser power (0.5 W) but without telling the laser spot size… We are used to measure on transparent materials with a power density below 100-1000 kW/m2 to secure absence of sample heating in cryostats without exchange gas. This limit of course depends a lot on the material itself but could the authors state about power density in their experiment? And at least give this value?
Response 2:
The pump power density can be estimated from the description of the experimental setup that is presented in the manuscript (lines 46, 47, 55). The laser power was 0.5W. The pump radiation was focused by the lens with focal length 35 cm. Commercial Ar-ion laser has an angular divergence of about 0.5 mRad. Simple calculations give the value of the pump power density at the sample to be 1.6kW/cm2. The sample of SrWO4 crystal was transparent.
Besides we checked heating of the sample by comparison its temperature when the pump radiation was ON and OFF. The temperature changed within ±1 K. This accuracy is noticed in the text.
Point 3:
Finally, in the ref. 15 an additional temperature at 295K was measured. Is there any reason why it is absent in Fig. 3 ?
Response 3:
We added the room temperature (T=295 K) experimental points in Fig. 3.
Reviewer 4 Report
The authors have addressed my questions. The manuscript can now be accepted.
Author Response
Response to Reviewer 4 Comments
We appreciate the reviewers’ time and constructive comments on our manuscript.
Following the suggestions, we have further improved our manuscript as highlighted in the text. Our responses to the single comments are given as follows:
Point 1:
The authors have addressed my questions. The manuscript can now be accepted.
Response 1:
Thank you for your positive decision.